# Resolving nanostructure and chemistry of solid-electrolyte interphase on lithium anodes by depth-sensitive plasmon-enhanced Raman spectroscopy

Yu Gu [1,5], En-Ming You [1,5], Jian-De Lin[1], Jun-Hao Wang[1], Si-Heng Luo[1], Ru-Yu Zhou[1], Chen-Jie Zhang[2], Jian-Lin Yao[2], Hui-Yang Li[1], Gen Li [1], Wei-Wei Wang[1], Yu Qiao [1], Jia-Wei Yan [1], De-Yin Wu [1], Guo-Kun Liu[3], Li Zhang[1], Jian-Feng Li [1], Rong Xu[4], Zhong-Qun Tian [1] ✉, Yi Cui [4] ✉ & Bing-Wei Mao[1] ✉

The solid-electrolyte interphase (SEI) plays crucial roles for the reversible operation of lithium metal batteries. However, fundamental understanding of the mechanisms of SEI formation and evolution is still limited. Herein, we develop a depth-sensitive plasmon-enhanced Raman spectroscopy (DS-PERS) method to enable in-situ and nondestructive characterization of the nanostructure and chemistry of SEI, based on synergistic enhancements of localized surface plasmons from nanostructured Cu, shell-isolated Au nanoparticles and Li deposits at different depths. We monitor the sequential formation of SEI in both ether-based and carbonate-based dual-salt electrolytes on a Cu current collector and then on freshly deposited Li, with dramatic chemical reconstruction. The molecular-level insights from the DS-PERS study unravel the profound influences of Li in modifying SEI formation and in turn the roles of SEI in regulating the Li-ion desolvation and the subsequent Li deposition at SEI-coupled interfaces. Last, we develop a cycling protocol that promotes a favorable direct SEI formation route, which significantly enhances the performance of anode-free Li metal batteries.

Lithium metal is regarded as an ideal anode for rechargeable Li batteries because of its ultrahigh specific capacity[1–3]. However, the electrochemical plating/stripping of Li metal usually presents a dendritic morphology, leading to low Coulombic efficiency and poor cycling stability of the Li metal anode[4–6]. The solid-electrolyte interphase (SEI), a nanometer-thin layer formed at the electrode-electrolyte interface due to the electrolyte decomposition, plays a crucial role in determining the Li deposition/dissolution and thus the reversible operation of Li metal batteries[7,8]. However, fundamental understanding of the SEI formation and evolution during battery operation is still limited, which limits the development of high-performance Li metal batteries.

The SEIs formed during the battery operation usually present structural and compositional heterogeneity – nanometric phases of decomposition products from the electrolyte are dynamically

[1]State Key Laboratory of Physical Chemistry of Solid Surfaces, iChEM, Innovation Laboratory for Sciences and Technologies of Energy Materials of Fujian Province (IKKEM), College of Chemistry and Chemical Engineering, Xiamen University, Xiamen, China. [2]College of Chemistry, Chemical Engineering and Materials Science, Soochow University, Suzhou, China. [3]State Key Laboratory of Marine Environmental Science, College of the Environment and Ecology, Xiamen University, Xiamen, China. [4]Department of Materials Science and Engineering, Stanford University, Stanford, CA, USA. [5]These authors contributed equally: Yu Gu, En-Ming You. ✉e-mail: zqtian@xmu.edu.cn; yicui@stanford.edu; bwmao@xmu.edu.cn

precipitated and distributed heterogeneously in the SEI[7,9]. The formation and evolution of SEIs become more elusive when the anode-free configuration of Li metal batteries is employed[10–13], in which SEI with different components and structures needs to form sequentially on the surface of the Cu current collector and then on the deposited Li[14–16]. Although SEIs have been intensively investigated by various advanced experimental techniques[17–26] and theoretical simulations[27,28], the preceding SEI formation on the Cu in the anode-free configuration and its possible detrimental effects on Li deposition-dissolution cycling stability is often overlooked. Therefore, in-situ methods that can follow dynamic interfacial processes and provide accurate molecular-level information is urgently needed to gain more detailed insights into the complicated interfacial processes involving SEI formation and evolution and to guide the design of favorable SEIs to improve the performance of Li metal batteries.

Surface-enhanced Raman spectroscopy (SERS) is a surface-sensitive technique based on the effect of localized surface plasmons (LSPs) of especially free-electron metal (e.g., Au, Ag, Cu and Li) nanostructures[29–31]. It is able to characterize the interfacial processes with molecular-level fingerprint information over a wide spectral range[32–35], thereby showing great potential to identify the composition and structure of SEIs in batteries[36–43]. However, since the LSP enhancement exhibit an exponential decay of electromagnetic intensity within a couple of nanometres[32], its application in characterizing the SEI with a thickness of tens of nanometers is limited – only the information from the vicinity of hotspots with the large local electromagnetic field can be extracted by SERS (see Supplementary Note 1). Alternatively, shell-isolated nanoparticle-enhanced Raman spectroscopy (SHINERS) is developed, in which plasmonic metal cores are coated with ultrathin inert shells to separate the cores from the substrate and are used solely as Raman signal amplifiers, giving reliable signals from the sample surface[44–51]. Monitoring of dynamic nature of SEI formation on anodes in Li-ion batteries via SHINERS has been reported, including a recent study involving Si anodes[52]. However, using the SHINERS to characterize the SEIs in Li metal batteries has rarely been reported[53], mainly due to the complex operation of SHINERS at the highly reactive Li metal surfaces and in organic electrolytes.

In this work, we develop a depth-sensitive plasmon-enhanced Raman spectroscopy (DS-PERS) method that combines the SERS and SHINERS to in-situ characterize the dynamic processes of SEI formation and evolution during the operation of Li metal batteries. The synergistic plasmonic enhancement of nanostructured Cu, shell-isolated Au nanoparticles, and Li deposits enables a depth-sensitive detection of signals from the SEI with a thickness of tens of nanometers and its interfaces to electrode (Cu or Li) and electrolyte (Fig. 1a). We further elucidate the formation of SEIs through two different routes, the sequential formation and the direct formation, highly depending on the cycling protocol. This comprehensive molecular-level information from the DS-PERS enriches our understanding of SEI formation and evolution during Li plating. Based on this understanding, we further design the favorable SEIs for practical anode-free Li metal using a potentiostatic-galvanostatic polarization strategy. The electrodes engineered with this type of SEI exhibited significantly enhanced cycling stability and an expanded lifetime in the anode-free Li metal batteries. The DS-PERS method provides tremendous opportunities for nondestructively characterizing the nanoscale interphases/interfaces with spatial information, which is a great challenge not only in the battery field but also in the general fields of material sciences and energy sciences.

## Results and Discussion
### Working principle of DS-PERS
The DS-PERS is based on a SERS-SHINERS integrated plasmonic enhancement structure with unique depth sensitivity by employing a SERS-active nanostructured Cu substrate, and shell-isolated nanoparticles (SHINs) introduced initially on top of the Cu substrate surface, as illustrated in Fig. 1a and Supplementary Notes 2–5. The SHINs had plasmonic Au cores (~60 nm) with ultrathin and pinhole-free SiO₂ shells (~2 nm), and the nanostructured Cu was created by the electrochemical oxidation-reduction roughening method[54] (Supplementary Figs. 1, 2). Before Li deposition, the junctions of adjacent nanostructured Cu serve as hotspots for providing the finger-print information of SEI formed in the vicinity of Cu surface (i.e., the inner region of SEI) in the early stage of formation (Fig. 1b). The LSPs-active SHINs provide additional signals from the hotspots of the SHINs and Cu such that the chemical and structural information of SEIs at outer regions can be obtained (Fig. 1c). After Li deposition, the Li metal with SHINs generated new hotspots (Fig. 1d) to further enhance Raman

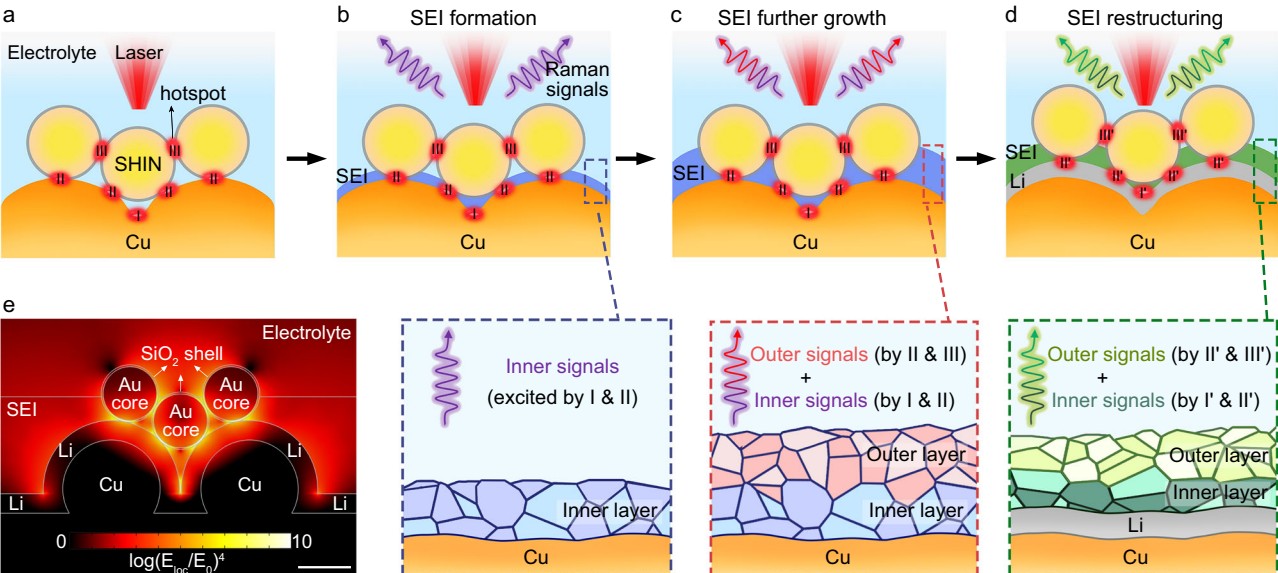

**Fig. 1 | DS-PERS by synergistic LSP enhancement pertinent to SEIs.**
**a–d** Schematics of the DS-PERS that enables detection of the signals from different depths in the SEI. The plasmonic structure of nanostructured Cu, SHINs and deposited Li generates strong electromagnetic field to enhance the Raman signals from the SEI. **e** The distribution of simulated electromagnetic field near the integrated plasmonic substrates during Li deposition. $E_{loc}$ and $E_0$ represent the localized field and the incident field, respectively. Scale bar: 60 nm.

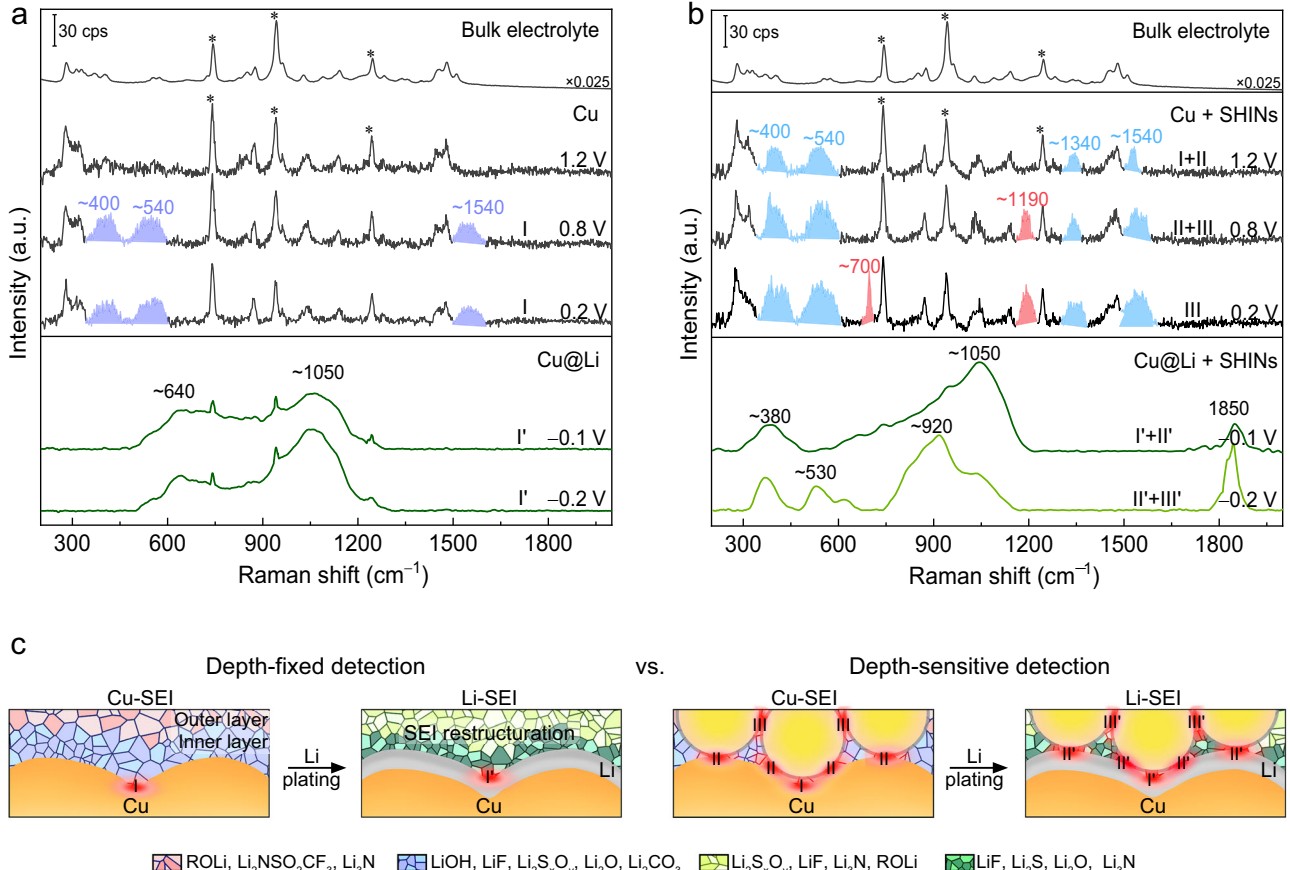

**Fig. 2 | Characterizations of SEI formed in the 1 M LiTFSI in DME-DOL using DS-PERS. a, b** In-situ Raman spectra showing the sequential formation and evolution of SEIs on the nanostructured Cu substrate (**a**) and the Cu-SHINs substrates (**b**) before and after Li deposition. Raman spectra of the bulk electrolyte are displayed for comparison. Peaks marked by asterisks are from the electrolyte. **c** Schematic comparison of the depth-fixed strategy based on LSPs of Cu alone and the depth-sensitive strategy based on synergetic LSPs of integrated Cu-SHINs for in-situ probing the formation and restructuring of SEI. By the DS-PERS, it is revealed that the primary Cu-SEI can undergo restructuring with the participation of freshly deposited metallic Li to form low-oxidation-state species with highly amorphous and heterogeneous structure.

signals of SEI near the Li surface and in the outer region by plasmonic coupling. The key advantage of such an elaborative design is the synergistic plasmonic enhancement effect, which provides higher detection sensitivity and depth sensitivity of Raman signals from SEIs and related interfaces. In this circumstance, we can capture the depth-dependent nanostructure and chemistry of SEI during its formation and evolution at different stages of Li plating.

The synergetic LSP enhancement was modeled using the finite element method (Fig. 1e and Supplementary Figs. 2–4 and Supplementary Table 1). For the Cu-SHINs integrated substrate (Supplementary Note 3 and Supplementary Fig. 3), the hotspots were formed in three types of junctions, namely, Cu island-to-Cu island (type-I), Cu island-to-SHIN (type-II) and SHIN-to-SHIN (type-III) junctions. The enhancement factor (EF) of such a configuration can reach up to $10^{10}$ (Supplementary Table 1), sufficiently high for the ultrasensitive detection of the Raman signals from the SEI. However, the distribution of SHINs on the Cu surface changed after primary SEI formation and Li deposition (Supplementary Fig. 4), and we need to consider how the change of SHINs locations influenced the EF. First, SEI formation was initiated only at the metal surface, leaving SHINs at least partially embedded in the SEI (Supplementary Note 4). However, since SEI components absorb light weakly and their dielectric constants are close to the electrolyte, the SEI growth will not substantially decrease the strength of the electromagnetic field from the plasmonic substrates (Supplementary Note 4). Even when the SHINs were entirely embedded in the SEI, the average EF was still as high as

$10^{10}$ (Supplementary Figs. 5, 6 and Supplementary Table 1). Second, two SHINs locations, on top of and within the Li deposit, were observed upon Li deposition. Only the SHINs on the Li deposits could generate effective hotspots to maintain a high EF of up to $10^{10}$ (Supplementary Note 5 and Supplementary Fig. 7). The above proposed plasmonic enhancement strategy with multiple hotspots provides a basis for the sensitive detection of Raman signals of the SEI during its formation.

To validate the synergetic LSP enhancement, we first performed the in-situ Raman measurements on the Cu electrode in an ether-based electrolyte of 1 M lithium bis(trifluoromethanesulfonyl)imide in 1,3-dioxolane and 1,2-dimethoxyethane (LiTFSI in DME-DOL)[1,55]. The Cu working electrodes with and without SHINs were assembled in a sealed three-electrode Raman cell where Li foils served as both the reference and counter electrodes (Supplementary Fig. 8a). A thin liquid layer (< 100 μm) was trapped between the Cu surface and quartz window to ensure that the Raman signals were from the metal-SEI interface, instead of the bulk electrolyte (Supplementary Fig. 8b). This was confirmed by the potential-dependent inversion of the intensity ratio of two prominent bands at 743 and 942 cm$^{-1}$, which are from the vibration modes of the TFSI anion (S–N and C–S stretching, CF$_3$ bending) and DOL solvent (C–O and C–C stretching), respectively (Fig. 2a, b, and Supplementary Figs. 11, 12).

We then conducted the potential-dependent Raman measurements to investigate the electrolyte reduction on a nanostructured Cu substrate (Fig. 2a and Supplementary Fig. 11). In the first cycle from 2.0

to 0.2 V without Li overpotential deposition (OPD), the cyclic voltammograms (CVs) of the Cu electrode show two successive cathodic waves starting from ~1.6 V (Supplementary Fig. 10) due to the solvent reduction and anion reduction, respectively[56]. However, the bands at ~400, ~540 and ~1540 cm$^{-1}$ for the reduction products of LiOH/LiF/ Li$_2$S$_x$O$_y$, ROLi/Li$_2$O and Li$_2$CO$_3$/ROCO$_2$Li (Supplementary Table 2), respectively, were not observed until the potential reached 1.0 V. The spectra remained the same upon a further decrease of potential to 0.2 V (Fig. 2a and Supplementary Fig. 11), even with SEI continual growth on the Cu surface. It means that Cu alone, due to the limited detection sensitivity of hotspot I, can be used only to probe the fingerprint information of the inner region of the SEI during its initial formation (Fig. 2c). Notably, bands for electrolyte components at the metal-electrolyte interface were still present even after the potential decreased to 0 V (Supplementary Fig. 11), likely due to a small number of electrolyte molecules buried in the SEI, which was recently observed by Cui and coworkers[57].

When the SHINs were integrated into the nanostructured Cu substrate (Fig. 2b and Supplementary Fig. 12), the Raman bands associated with the electrolyte reduction appeared as early as 1.4 V (Supplementary Fig. 12), indicating the higher detection sensitivity of the Cu-SHINs substrate with multiple hotspots (hotspots I and II). When the potential is lower than 1.2 V, new bands at 1340, 1191, and 700 cm$^{-1}$ (excited by hotspots II and III) appeared sequentially (Fig. 2b and Supplementary Fig. 12), which were attributed to the formation of ROLi/Li$_2$S$_x$O$_y$, Li$_2$NSO$_2$CF$_3$/Li$_2$S$_x$O$_y$ and Li$_3$N, respectively, from the successive reductions of TFSI and DOL (Supplementary Table 2). It verified that the synergistic LSPR enhancement from the integrated Cu-SHINs substrate enabled Raman signals to be detected from different depths of the SEI during its formation. Since the SHINs were initially located on the top of the Cu surface and later presented in the outer region of the growing SEI, this observation serves as convincing evidence for the outer surface growth mechanism of SEI (Fig. 2c).

Pronounced changes in Raman spectra were observed on both the nanostructured Cu and integrated Cu-SHINs substrates during a negative potential excursion to −0.1 V where the Li OPD took place (lower panels in Fig. 2a, b). The Raman bands of SEI species observed at the positive potential disappeared, while strong and broad bands overlapped in some specific regions for the SEIs formed on the Cu substrate and over the entire wavenumber range for the SEIs formed on the integrated Cu-SHINs substrate. These phenomena suggest that the SEI was undergoing reconstruction with the participation of freshly deposited Li (Supplementary Note 6). The largely broadened shape of the vibrational bands indicates that the SEI on the Li surface is highly amorphous and heterogeneous. Upon further Li deposition, significant changes in the Raman spectra were observed only for the SEI on the Cu-SHINs substrate (lower panels in Fig. 2a, b), again demonstrating that the synergistic plasmonic substrates can create significant Raman signal enhancement and thus enable depth-sensitive detection of SEI.

To study the reconstruction of SEI during Li deposition, we conducted the in-situ Raman measurements using precise control of applied potential at an interval of 5 mV (Supplementary Fig. 13). The resulting broad bands were deconvoluted for component analysis (Supplementary Fig. 14 and Supplementary Table 3). Prior to Li OPD, the SEI formed on the Cu surface displayed a typical organic-inorganic hybrid structure, in which high- and low-oxidation-state species (e.g., ROLi, Li$_2$S$_x$O$_y$ and LiF) were the major components of the SEI. However, after the Li OPD was initiated, both the inner and outer regions of the SEI underwent reconstruction with higher-oxidation-state species decomposed to lower-oxidation-state species (e.g., Li$_2$O, LiF, Li$_2$S and Li$_3$N). The component analysis was supported by XPS profile analysis (Supplementary Fig. 15). We expect that the role of metallic Li is to promote additional chemical reactions to significantly alter the structure of the primary SEI on the Cu current collector to the final SEIs on the deposited Li (Supplementary Note 6 and Supplementary Fig. 16).

Such a sequential formation and reconstruction of SEIs are overlooked in the past research on Li metal batteries.

## Sequential and direct formation of SEIs in carbonate-based electrolytes

The DS-PERS method was then used to study the formation and evolution of SEIs in a complex yet promising carbonate-based dual-salt electrolyte of lithium difluoro(oxalato)borate and lithium tetrafluoroborate in diethyl carbonate and fluoroethylene carbonate (LiDFOB-LiBF$_4$/DEC-FEC) proposed by Dahn and coworkers[58,59]. We apply a galvanostatic polarization (denoted as the G route) to a Cu||Li half-cell, where an integrated Cu-SHINs plasmonic substrate was used as the working electrode. The resulting potential profiles and time-dependent Raman spectra were recorded simultaneously in Fig. 3a, b.

As shown in Fig. 3a (black line), the potential on the Cu electrode was decreased to around 0 V within the first 100 s, during which the electrolyte was reduced to the primary SEI on Cu (denoted as Cu-SEI) without the participation of Li OPD. In the corresponding Raman spectra (Fig. 3b and Supplementary Fig. 18), broad background-like features in the regions of 300–600 cm$^{-1}$ and 1200–1600 cm$^{-1}$ gradually appeared due to the B-containing decomposition products of LiDFOB salt and the carbonyl species from the reduction of DEC-FEC solvents (Supplementary Table 4). At ~90 s, a new band at ~1011 cm$^{-1}$ appeared from the B−O/C−O stretching modes of B-containing components (Supplementary Fig. 18), implying that the primary SEI formed on the Cu surface was organic-rich.

As the polarization continued, the potential of the Cu electrode became slightly lower than 0 V, implying the occurrence of Li OPD (Fig. 3a, red line). Accordingly, dramatic changes in the Raman spectra were observed: bands initially seen when the potential is above 0 V disappeared, while four new broad bands associated with organic B-containing species (at ~370 and ~540 cm$^{-1}$) and Li carbonates/alkoxides (at ~1300 and ~1490 cm$^{-1}$) appeared (Fig. 3b, Supplementary Fig. 18 and Supplementary Table 4). These species were distributed successively from the inner to outer regions of the SEI, as evidenced by comparative Raman spectra recorded on the Cu-SHINs and bare nanostructured Cu substrates (Supplementary Fig. 19). The intensities of these bands continued to increase over time until after 30 min, indicating that the SEI formation was completed. Deconvolution analysis (Supplementary Fig. 23 and Supplementary Table 4) suggested that the final SEI was enriched by a mixture of organic and inorganic species (e.g., B−F compounds and Li$_2$CO$_3$) throughout the thickness of the SEI. Some of the mixture species were still in high-oxidation states. We conclude that under the galvanostatic polarization, the SEI was sequentially formed by electrolyte reduction at above 0 V (prior to Li OPD), and followed by further electrolyte reduction below 0 V (with Li OPD). This transition from the primary SEI on the Cu current collector (i.e., Cu-SEI) to the final SEI on the Li (i.e., Li-SEI) was accompanied by a significant chemical reconstruction, as schematically shown in Fig. 3c.

The composition and structure of the final Li-SEI were closely related to the potential of electrolyte reduction. We found that prolonging the bias at the potential prior to Li OPD can promote the enrichment of high-oxidation-state organic components of the final SEI, especially in the outer region, as confirmed by potential-dependent Raman measurements (Supplementary Fig. 20 and Supplementary Table 5). However, when the potential was decreased rapidly to below 0 V, the formation of the primary Cu-SEI was suppressed, leading to the direct formation of the SEI on the Li. Herein, we design an electrochemically engineered strategy with a potentiostatic-galvanostatic polarization (P-G route) to artificially construct a favorable SEI. The P-G route initiate with a negative potentiostatic stepping (−0.1 V for 100 s, blue region in Fig. 3d) to incur the Li OPD and electrolyte reduction for a certain time, followed by a galvanostatic plating (-0.5 mA cm$^{-2}$, purple region in Fig. 3d) to stabilized SEI formation. Broad Raman bands characteristic of Li-SEI appeared immediately

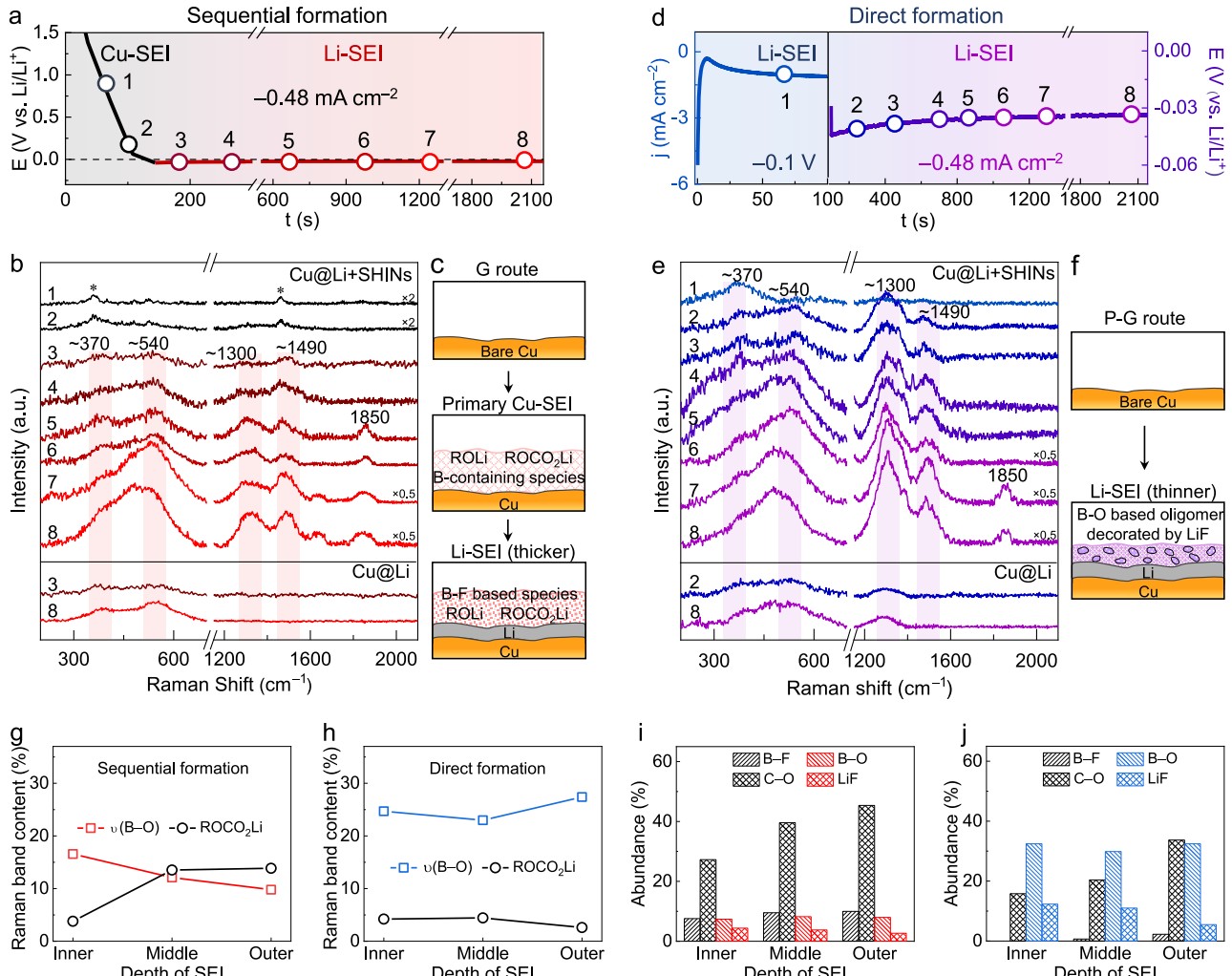

**Fig. 3 | Formation and evolution of SEIs in the dual-salt electrolyte revealed by DS-PERS. a–c** $E–t$ curve of the galvanostatic polarization (denoted as the G route) of Li plating on Cu (**a**). In-situ Raman spectra recorded during the formation and evolution of the SEI under the G route (**b**). Peaks marked by asterisks are from the electrolyte. Schematics of the structure and chemistry of the SEI formed under the G route (**c**). **d, e** $j–t$ and $E–t$ curves of electrochemically engineered potentiostatic-galvanostatic polarization (denoted as the P-G route) of Li plating on Cu (**d**). In-situ Raman spectra recorded during the formation and evolution of the SEI under the P-G route (**e**). Schematics of the structure and chemistry of the SEI formed under the P-G route (**f**). **g, h** Normalized Raman peak area of bands associated with B–O and carbonate species in sequentially formed Li-SEI (**g**) and directly formed Li-SEI (**h**). **i, j** Percentages of specified components in the sequentially formed Li-SEI (**i**) and directly formed Li-SEI (**j**), calculated from the XPS spectra.

upon the cathodic potential stepping due to the absence of Cu-SEI (Fig. 3e and Supplementary Figs. 21, 22). During the following galvanostatic plating, the intensities of the broad bands increased markedly and approached saturation after only ~15 min, indicating faster evolution of SEI than the G route. Importantly, the chemistry and structure of the SEI were significantly different from the sequentially formed SEI under the G route (Fig. 3f), as further revealed by the deconvoluted spectra in Supplementary Fig. 23 and Supplementary Table 4.

The sequential formation of SEIs under the G route and the direct formation of SEI under the P-G route were further investigated using XPS characterization and density functional theory (DFT) calculations (Supplementary Figs. 24–26 and Supplementary Table 4). We found that the broad bands appeared in both routes due to the wide distribution of size and microenvironment of the highly amorphous Li-SEI. Nevertheless, the directly formed Li-SEI by the P-G route had a notably reduced thickness compared to the sequentially formed Li-SEI from the primary Cu-SEI (Supplementary Fig. 24). Moreover, the organic B–F components from the decomposition of LiDFOB were present in the sequentially formed SEI; such components were hybridized with inorganic species of LiF with varying amounts across the

SEI, as evidenced by the XPS depth profile (Supplementary Fig. 25). Under the P-G route, since the electrolyte experience a large driving force for reduction at the potential below 0 V, DFOB anions can capture electrons from Li metal to enable the B–O or B–F bond cleavage; thus low-oxidation-state inorganic species such as LiF (Supplementary Fig. 25) became the main components throughout the Li-SEI. More importantly, the sequentially formed Li-SEI contains more organic species, especially in the outer region occupied by high-oxidation-state organic species such as ROCO$_2$Li and ROLi (Fig. 3g, i). However, the directly formed Li-SEI exhibited a distinctive polymeric-like structure. The vibrations of O–B–O/CH$_2$ at 1295 cm$^{-1}$ and Li–O/(LiF)$_n$ at 470 cm$^{-1}$ from crosslinked oligomeric borates (e.g., –B(OCH$_2$CH$_2$)$_3$–) became intensified while carbonate species became weakened (Fig. 3h, j and Supplementary Fig. 25). Reactions of the electron-deficient B-containing intermediates from DFOB anion decomposition with other SEI components (e.g., electron-rich lithium alkoxide or lithium semicarbonates) could be one of the reasons that leads to the crosslinking (Supplementary Fig. 26). It is generally acknowledged that the SEIs with polymeric-like structures are beneficial to suppressing Li dendrite growth[60,61]. Such SEIs composed of oligomers incorporated

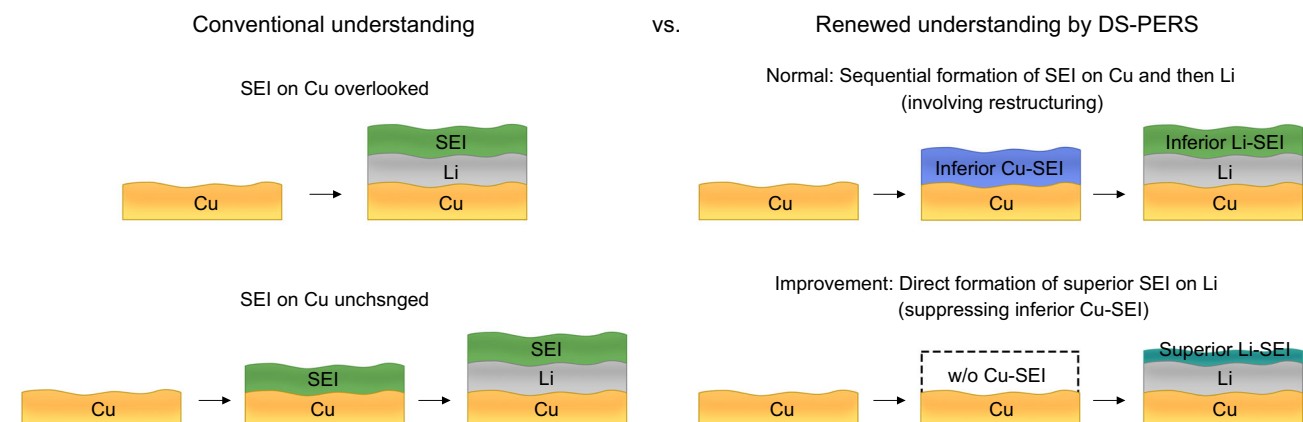

**Fig. 4 | Schematics of conventional and renewed understandings of SEIs formation and evolution.** In the conventional understanding, either the SEI formation on Cu is overlooked, or the SEI formed on Cu is considered to remain unchanged after Li deposition. The DS-PERS revealed a sequential formation of SEI on the Cu and then on the Li, with a significant chemical and structural reconstruction.

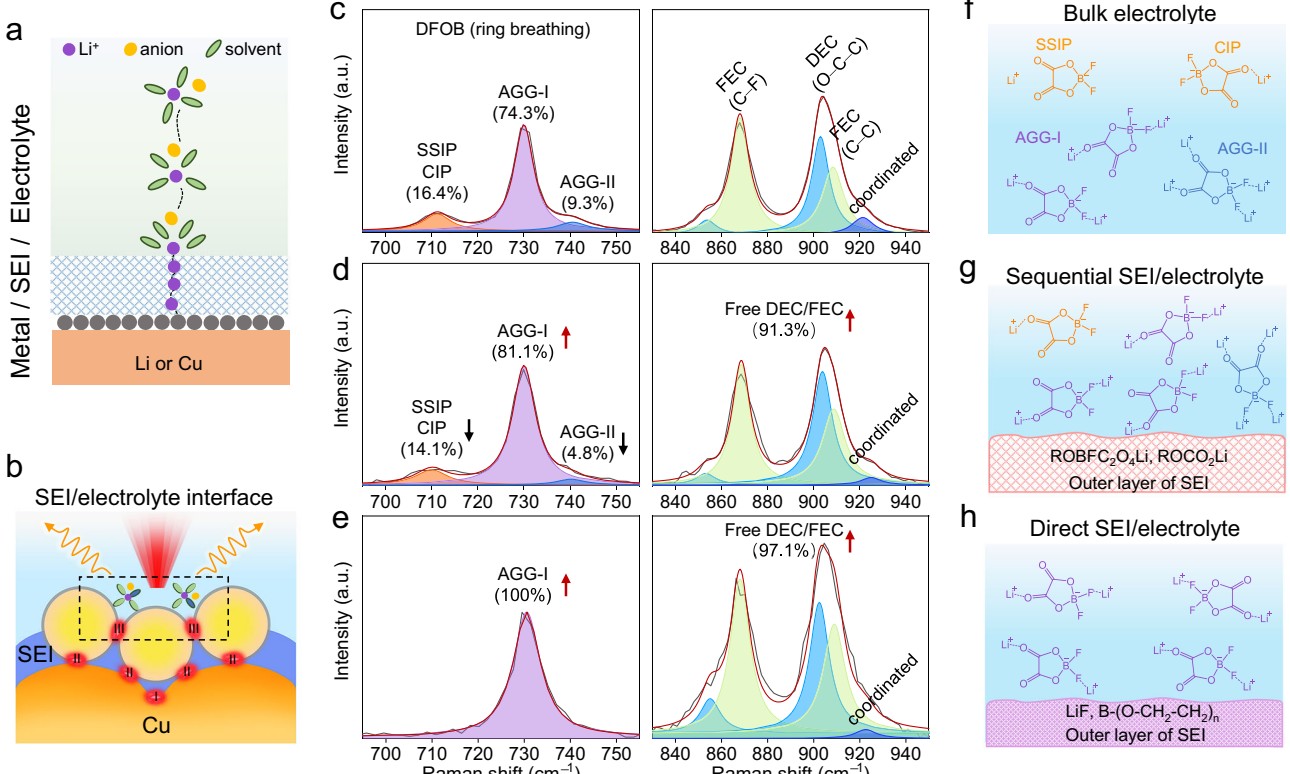

**Fig. 5 | The desolvation of Li-ions at different SEI/electrolyte interfaces revealed by DS-PERS. a** Schematic for Li electrodeposition on the electrode with the SEI. The solvated ions in the electrolyte need to be desolvated at the SEI/electrolyte interface and then transported through the SEI to the metal/SEI interfaces for reactions. **b** Schematic the schematic of the DS-PERS that can in-situ collect the signals from the SEI/electrolyte interface. The hotspots III enhance the Raman signals from the SEI/electrolyte interface. **c–e** Potential-dependent Raman spectra of Li-ion solvation structure at bulk electrolyte (**c**), sequentially-formed SEI/electrolyte interface (**d**) and directly-formed SEI /electrolyte interface (**e**). **f–h** Schematics of Li-ion solvation structure governed by the chemical structure of SEIs.

with inorganic species are usually mechanically robust and more facile for Li-ion transport. The above information demonstrates that the DS-PERS with synergetic LSP enhancements is powerful in elucidating the different SEI formation mechanisms, and such in-depth understanding can be truly rewarded by designing the favorable SEI to enable the stable cycling of Li metal batteries (Fig. 4).

## Desolvation of Li-ion at SEI/electrolyte interface

The desolvation of Li ions starts at the SEI/electrolyte interface, followed by the Li-ion transport through the SEI to reach the metal/SEI

interface for charge transfer reactions (Fig. 5a). To reveal the roles of the SEI/electrolyte interface in regulating the desolvation of Li-ion and follow-up Li electrodeposition, DS-PERS was utilized to collect the enhanced Raman signals from the hotspots III at the SEI/electrolyte interface (Fig. 5b).

The Raman spectra of bulk electrolyte (LiDFOB-LiBF$_4$ in DEC-FEC) are shown in Fig. 5c (left panel). The DFOB anion bands at 700 ~ 750 cm$^{-1}$ can be deconvoluted into three distinctive bands at 711, 730 and 741 cm$^{-1}$ associated with solvent-separated ion pairs (SSIPs, non-coordinated DFOB)/contact ion pairs (CIPs, DFOB coordinating to

one Li-ion), aggregates-I (AGGs-I, DFOB coordinating to two Li-ions) and AGGs-II (DFOB coordinating to four Li-ions), respectively (Supplementary Fig. 27). Obviously, the majority of DFOB anions exists as AGGs-I with small amounts of SSIPs/CIPs and AGGs-II due to the strong interaction between DFOB and Li$^+$ (Fig. 5f). This is also verified by the spectra of solvents (Fig. 5c, right panel), where Raman bands from fewer coordinated solvents to Li$^+$ are observed.

For the interface between the sequentially formed SEI and electrolyte (Fig. 5d), the local solvation structure is different from that in the bulk electrolyte, where the number of free DFOB anions reduces to form more AGGs through interactions with multiple Li-ions. We expect that the outer SEI region consisting of electron-rich species such as ROCO$_2$Li and ROBFC$_2$O$_4$Li can repel the negatively charged DFOB anions, leading to the coexistence of SSIPs/CIPs and AGGs at this interface. However, at the interface between the directly formed SEI and electrolyte (Fig. 5e), the solvation structure is dominated by AGGs-I, as reflected by the increase of the intensity of free solvents (largely weakened interaction with Li$^+$). In this circumstance, inorganic LiF and crosslinked oligomeric Li−B(OCH$_2$CH$_2$)$_3$−R are the major components in the outer region. These species have higher interfacial energy that may modify the Li$^+$ solvation environment through the interfacial interactions. For example, the LiF tends to bind with C = O and B−F in the DFOB anion. DFT calculations show the large binding energies of the LiF coordinated with C = O (−0.33 eV) and the LiF coordinated with both C = O and B−F (−0.39 eV), leading to the significant increase in the proportion of AGGs-I that has spare C = O and B−F coordination bonds to LiF. In summary, the structure and chemistry of SEI, especially in the outer region, can dramatically affect the Li$^+$-anion interaction at the SEI/electrolyte interface. The change in the solvation structure of Li-ions is anticipated to have a significant impact on its desolvation behavior and consequently alter the Li deposition. It demonstrates again that the DS-PERS can serve as a powerful tool to reveal the important roles of SEI/electrolyte interfaces in the Li-ion desolvation and Li deposition on the Li metal anode.

**Improve battery performance with the designed favorable SEIs**

We expect that the directly formed SEI and its interfaces are favorable for the stable Li deposition/dissolution on the Li metal anode. The improved performance of the electrode with such an SEI is demonstrated in Fig. 6a–d, where the performance of Cu||Li cells with Cu current collectors covered by different SEIs are compared. The electrolyte used here is the LiDFOB-LiBF$_4$ in DEC-FEC.

The cells with directly formed Li-SEI exhibited a long cycle life of over 200 cycles and an average Coulombic efficiency (CE) as high as 99.5% (Fig. 6a). In addition, the Li deposition overpotential remained stable without presenting an obvious voltage hysteresis (Fig. 6c, d). In contrast, the one with the sequentially formed SEI showed a much shorter cycle life (Fig. 6a) and dramatically varied voltage hysteresis during cycling (Fig. 6b). Moreover, with directly formed Li-SEI, a uniform Li deposit was formed without apparent dendrite and porous structures (Fig. 6e, f). The cell performance can be further improved by the periodic application of a higher current density pulse of optimized magnitude during charge-discharge cycling, leading to the rapid deposition of Li metal to maintain the high quality of directly formed SEI. It can be obviously observed that the cell with in-situ restoring showed more stable cycling with a further boosted CE of 99.6% (Fig. 6g, h).

We further assessed the potential of engineering the directly formed SEI to enable the stable cycling of anode-free full-cells. Proof-of-concept anode-free full cells using the commercial LiNi$_{0.5}$Mn$_{0.3}$Co$_{0.2}$O$_2$ (NMC532) cathode were tested (Fig. 6i, j). For the cell with the directly formed SEI on the Cu current collector, it delivered the initial discharge capacity of ~176 mAh g$^{-1}$, and approximately 80 cycles with a high and stable CE were achieved at a 0.2 C charge/0.5 C discharge rates before the capacity fell to 80%, as shown in Supplementary Fig. 28. For the cell

with the sequentially formed SEI, considerable performance degradation was observed. After 20 cycles, the morphology of deposited Li on the Cu with sequentially formed SEI was dendritic and porous (Fig. 6k). In contrast, the deposited Li on Cu with the directly formed SEI showed relatively densely packed and flat grains even after 50 cycles (Fig. 6l).

In summary, we developed a DS-PERS method for in-situ characterization of the formation and evolution of SEI in Li metal batteries. The synergistic LSP enhancement from nanostructured Cu, SHINs and Li deposits was utilized to enable bottom-up nondestructive detection of the structure and chemistry of SEIs and related interfaces with depth sensitivity. The dynamic molecular-level information of the SEI revealed that the sequential formation of SEIs on the bare Cu current collector and then freshly deposited Li is a universal phenomenon in both ether-based and carbonate-based electrolytes, which is overlooked in the conventional understanding of SEI formation. Specifically, the primary SEI formed on Cu is composed of less stable high-oxidation-state components and has to undergo chemical restructuring upon subsequent concurrent Li deposition to complete the formation of a final SEI on Li, which is less desirable for battery performance. Such a sequential SEI formation route can be altered by applying an electrochemical strategy to promote rapid Li deposition and thus suppress the preceding primary SEI formation on Cu, leading to the direct formation of SEIs on Li with the enrichment of more stable low-oxidation-state components. This in-depth understanding of SEI and related interfaces guided us to design a potentiostatic-galvanostatic polarization strategy for achieving an improved polymeric-like structured Li-SEI in a dual-salt carbonate-based electrolyte. The electrodes engineered with this type of SEI exhibited significantly enhanced cycling stability and an expanded lifetime in the anode-free Li metal batteries. This study expands the capability of DS-PERS to study the complicated interfaces/interphases in Li metal batteries, which opens up a new perspective in real-time investigation and understanding of the mechanisms of SEI formation and evolution and the roles of SEI in regulating Li-ion desolvation and Li deposition, which is pivotal in the rational construction of SEIs and related interfaces for practical batteries including other alkali metal batteries.

## Methods

### Preparation of nanostructured Cu substrate

The traditional oxidation-reduction cycle (ORC) method was used for electrochemical roughening of the surface of the Cu substrate to obtain nanostructures for effective SERS activation[54]. The ORC treatment was performed in a 0.1 M KCl aqueous solution using a potential step procedure described as follows: oxidation at 0.4 V for 5 s followed by reduction at −0.4 V for 5 s, which was repeated for 3 cycles. The obtained nanostructured Cu substrate was rinsed with deionized water and rapidly dried under vacuum conditions.

### Synthesis and assembly of Au@SiO$_2$ SHINs

Au@SiO$_2$ SHINs were synthesized following previous report[44]. Au nanospheres with a diameter of ~60 nm were first prepared using the sodium citrate reduction method. Then, 200 mL of HAuCl$_4$ aqueous solution (0.01 wt.%) was heated to boiling under string. A total of 1.6 mL of sodium citrate aqueous solution (1 wt.%) was quickly added to the above boiling solution and refluxed for 1 h. Then, the sol was allowed to cool to ambient conditions for the next step. The ultrathin SiO$_2$ shell was then coated by the high-temperature silicate hydrolyzation method. A 400 µL of (3-aminopropyl) trimethoxysilane (1 mM) was added to 30 mL of the as-prepared Au nanosphere sol under stirring without heat. After 15 min of stirring at room temperature, a sodium silicate solution with the pH adjusted to approximately 10.0 by H$_2$SO$_4$ was added and stirred for another 5 min. Then, the mixture was transferred to a boiling water bath and stirred for approximately 30 min to coat the pinhole-free SiO$_2$ shell with ca. 2 nm in thickness. The as-prepared Au@SiO$_2$ nanoparticles were centrifuged thrice and washed with deionized water for later use.

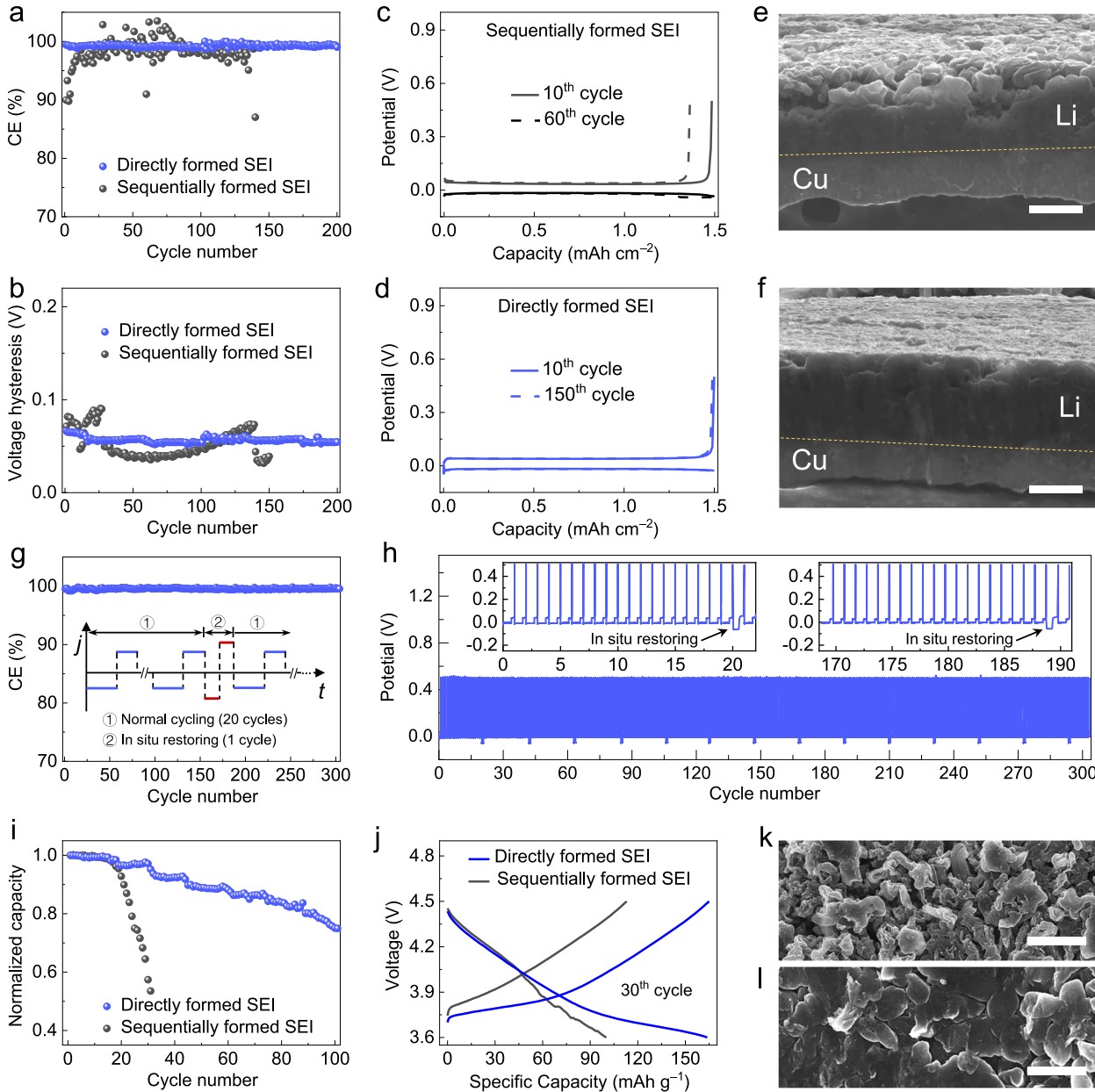

**Fig. 6 | Electrochemical performances of anode-free electrodes covered by different SEIs. a** CE of Li deposition/dissolution on Cu electrodes covered by different SEIs (deposition at 0.5 mA cm⁻² and dissolution at 1.25 mA cm⁻²; capacity of 1.5 mAh cm⁻²). **b** The hysteresis of Li deposition/stripping for the Cu electrodes covered by different SEIs. **c, d** Voltage profiles of the cells with Cu electrodes covered by sequentially formed SEI (**c**) and directly formed SEI (**d**). **e, f** Cross-sectional SEM images of Li deposited on the Cu electrodes covered by sequentially formed SEI (**e**) and directly formed SEI (**f**). Scale bar: 10 μm. **g, h** CE (**g**) and voltage profiles (**h**) of Li deposition/dissolution on Cu electrode covered by directly formed SEI with periodic in-situ restoring under higher current density (deposition at 5.0 mA cm⁻² and dissolution at 5.0 mA cm⁻²). **i, j** Capacity retention (**i**) and voltage profiles (**j**) of the coin-type Cu||NMC532 cells with Cu electrodes covered by different SEIs (charge at 0.2 C and discharge at 0.5 C). **k, l** SEM images of plated Li morphology in the Cu||NMC532 cells with Cu electrodes covered by sequentially formed SEI (**k**) and directly formed SEI (**l**), respectively. Scale bar: 10 μm.

For the assembly of Au@SiO₂ SHINs on the surface of the nanostructured Cu substrate, the above concentrated sol was diluted with 0.5 mL of deionized water to make up a stock solution. Two microliters of the stock solution were then drop-cast onto the surface of a freshly nanostructured Cu substrate and rapidly dried under vacuum overnight.

### Characterization of the samples

The morphologies of the nanostructured Cu surface and Au@SiO₂ SHINs were characterized by scanning electron microscopy (SEM, HITACHI S-4800) and high-resolution transmission electron microscopy (HRTEM, F30) coupled with energy dispersive X-ray spectrometry,

respectively. The distribution of Au@SiO₂ SHINs on the nanostructured Cu substrate after Li deposition as well as the morphology of the Li metal surfaces after cycling were characterized by SEM (Zeiss GeminiSEM 500) with an air-isolating transfer apparatus to avoid air exposure of the samples. XPS measurements were performed on an ESCALAB Xi+ (Thermo Scientific) spectrometer using monochromatic Al Kα (1486.7 eV) X-ray source at 15 KV with a beam spot size of 650 μm. The binding energies were referenced to the C 1s line at 284.6 eV from adventitious carbon. Depth profiling was fulfilled using Ar ion sputtering in the x-y scan mode at ion acceleration of a 2 kV and ion beam current of 2 μA over an area of 2 × 2 mm². The thickness of the SEI was estimated based on the calibrated sputtering rate of 5 nm per minute for Ta₂O₅.

## In-situ electrochemical Raman spectroscopy

In-situ Raman measurements were conducted on a Raman-11 system (Nanophoton) equipped with a 50× (NA 0.45) objective and a 300 grooves/mm grating. A 785 nm laser was used to avoid the fluorescence background from the electrolyte (Supplementary Fig. 9). The homemade sealed Raman cell, in which the nanostructured Cu decorated by SHINs served as the working electrode and Li foils served as both the reference and counter electrodes, was assembled in an Ar-filled glovebox (< 0.1 ppm of $H_2O$ and $O_2$). The working electrode surface faced upwards and towards a quartz glass window of the cell with 0.5 mm thickness. The laser power at electrode was approximately 0.7 mW $\mu m^{-2}$ and the acquisition time was 60 s for each spectrum. Raman frequencies were calibrated using a Si wafer. To remove the interference of fluorescence, the Raman spectra were background-corrected based on a fifth-order polynomial function. For a higher signal-to-intensity ratio, while preserving the spectroscopic features, a peak extraction and retention algorithm[62] was applied to the Raman spectra recorded on bare nanostructured Cu and integrated Cu-SHINs substrates after Li deposition, as shown in Fig. 2.

## Measurements of electrochemical performances

All electrochemical tests were carried out in a 2016-type coin-cell configuration and all cells were fabricated in an Ar-filled glovebox, with one layer of Celgard 2325 used as a separator and 50 µL electrolyte. Galvanostatic charge-discharge cycling was performed on LANHE CT2001A battery testing system. Two types of SEIs were preformed on Cu current collectors with Cu||Li half-cell configurations before electrochemical testing. The sequentially formed SEI on Cu (19 mm diameter) was prepared by normal galvanostatic route with cathodic polarization at 0.5 mA $cm^{-2}$ for 5 h followed by stripping to 0.5 V at 1.25 mA $cm^{-2}$. The directly formed SEI on Cu was prepared by potentiostatic-galvanostatic route, where a potential of −0.1 V was applied for 100 s in the potentiostatic period and a constant cathodic current density of 0.5 mA $cm^{-2}$ was immediately applied for 1 h in the subsequent galvanostatic period. The residual Li after the potentiostatic-galvanostatic route was removed by electrochemical dissolution. For Coulombic efficiency tests, Cu||Li half-cells were cycled by depositing 1.5 mAh $cm^{-2}$ of Li onto the Cu current collector with different preformed SEIs at −0.5 mA $cm^{-2}$ followed by stripping to 0.5 V at 1.25 mA $cm^{-2}$. The Coulombic efficiency was calculated by dividing the total stripping capacity by the total deposition capacity. For periodic in-situ restoring strategy, a higher current density pulse of 5 mA $cm^{-2}$ was applied to Cu||Li half-cell every 20 normal cycles. For anode-free full cell tests, Cu||NMC532 cells were assembled using a Cu current collector (19 mm diameter) with differently preformed SEIs as the anode and commercial NMC532 (Easpring) coated on Al foil (13 mm diameter) as the cathode (94.5 wt.% active). The typical loading of NMC532 was approximately 15 mg $cm^{-2}$. Before the measurements, all the cells were subjected to two formation cycles at C/10 rate for both charge and discharge processes. These cells were cycled between 3.6 V and 4.5 V at the C/5 rate for charge and the C/2 rate for discharge. 1 C is equal to 170 mA $g^{-1}$ of active NMC532 materials. For these conditions the areal capacity was about 2.5 mAh $cm^{-2}$. All electrolytes used were 0.6 M LiDFOB-0.6 M $LiBF_4$/DEC-FEC (2:1, V/V, anhydrous > 99%, DoDoChem), and all cells were tested at 25 °C. Each test was repeated with three coin cells to ensure consistency.

## Theoretical simulation and calculation

The simulated electromagnetic field was obtained by commercial simulation software (COMSOL Multiphysics) based on the finite element method. A spherical simulation domain, whose diameter was 1.8 µm, was created, and perfectly matched layers (PMLs) were employed to simulate an open boundary. The bottom half of the simulation domain was set as Cu. The medium over the Cu substrate was set to be the electrolyte or SEI. A quarter of the spherical Cu

nanoparticles, whose diameters were 57 or 100 nm, were buried in the Cu substrate. The size of the nanogap was 2 nm for the adjacent Cu nanoparticles. The Au core (55 nm)-$SiO_2$ shell (2 nm) nanoparticles were modelled as core-shell nanoparticles. The minimal mesh size was 0.1 nm in close proximity to the nanoparticles and gradually became coarser towards the borders of the simulation domain. The refractive indexes of the electrolyte and the SEI were 1.4 and 1.6, respectively[63]. The permittivity for Au, Cu, Li and $SiO_2$ were taken from references[64,65]. Density functional theory (DFT) calculations were implemented with the B3LYP functional[66,67] and 6−311 + G(d, p) basis set. The implicit solvent model (SMD)[68] was used with acetone (a dielectric constant of 20.5) as the default solvent to consider the role of solvent effect. The structures of FEC, DEC, LiDFOB and the corresponding reaction process as well as DFOB-Li$^+$ coordination were performed by using Gaussian 09 program package[69].

## Data availability

The data that support the findings of this study have been included in the main text and Supplementary Information. All other relevant data supporting the findings of this study are available from the corresponding authors upon request.

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

## Acknowledgements

This work was supported by the National Natural Science Foundation of China (22002129 (Y.G.), 21972119 (B.W.M.), 21991151 (B.W.M.), 22202162 (E.M.Y.), 22072123 (J.W.Y.), 22102137 (W.W.W.)), the China Postdoctoral Science Foundation (2019TQ0177 (Y.G.), 2022M722648 (E.M.Y.), 2022T150548 (W.W.W.)). We thank H.X. Lin, C.Y. Li, and F.R. Fan for helpful discussions and advice, Y.H. Hong for experimental assistance and C. Geng for help on schematic design.

## Author contributions

Z.Q.T., B.W.M., and Y.G. conceived idea. Y.G., B.W.M., and Z.Q.T. designed experiments and analyzed results. Y.G., E.M.Y., B.W.M., Z.Q.T, C.Y., and R.X. wrote the manuscript. Y.G., J.H.W., R.Y.Z, C.J.Z., G.L., H.Y.L and W.W.W. performed experiments. E.M.Y. conducted the COMSOL simulations. J.D.L. conducted the DFT calculations. S.H.L., J.L.Y., Y.Q., J.W.Y., D.Y.W., L.Z., G.K.L. and J.F.L. helped with discussion. All authors discussed and analyzed the data.

## Competing interests

The authors declare no competing interests.
