## [Peer Review File · Nature Communications]

Resolving nanostructure and chemistry of solid-electrolyte interphase on lithium anodes by depth-sensitive plasmon-enhanced Raman spectroscopyREVIEWER COMMENTS

Reviewer #1 (Remarks to the Author):

The work presented is a systematic study of the use of SHINERS and SERS to follow SEI formation on Cu substrates. A key outcome is the demonstration of discriminating depth and region of the SEI - by design and combination of the substrates.

Insights from the spectroscopy led to authors demonstrating direct improvement to the electrochemical properties of lithium plating and stripping on Cu - by identifying a less stable SEI formed on the copper and providing a method to address this.

Overall this is a well carried out study with important incite on the nature of the SEI formed upon Cu/Li.

The authors give a good level of citation of the past literature. They may wish to considering to add citations to earlier pioneering Raman studies on lithium metal - particularly work from Rey et al *Electrochimica Acta* 43, 1998, 1539-1544 and Naudin et al *Journal of Power Sources* 124, 2003 518-525 (paper ends with a pertinent remark - "However, the sensitivity of normal Raman is rather limited and SERS effects would be very helpful to characterize the SEI") 20 years later this is beginning to be realised, and the SHINERS study on lithium metal from Galloway at al. *Faraday Discuss.*, 2017, 205, 469-490.

On page 7 the authors state that "such a sequential formation and reconstruction of SEIs are overlooked in past research". A recent study from Unwin and co-workers (*Angewandte Chemie International Edition* 61, e202207184 2022) shows that monitoring SEI formation on Si via SHINERS shows the dynamic nature of SEI formation as well. Agree overlooked, but studies are appearing that begin to identify the changing nature of SEIs which this work adds to.

Reviewer #2 (Remarks to the Author):

The manuscript titled "Resolving the nanostructure and chemistry of solid-electrolyte interphase in lithium metal batteries by depth-sensitive plasmon-enhanced Raman spectroscopy" proposed a new method of probing the solid/electrolyte interphase on Li anode by using shell-isolated Au nanoparticles SERS media. The major finding is that the SEI evolution on the Cu and on plated Li can be sequentially explored. Insights learned from this method benefit optimization of the electrochemical cycling of the Li metal battery to generate a more stable SEI to promote cell performance. Overall, it is nice work and should attract the broad interest of the researchers working in the Li-ion battery field. Before it can be published in *Nature Communications*, the following comments should be properly addressed:

A. Regarding the principle of DS-PERS described in Figure 1:

1. Early formation of SEI is probed by the roughened surface of Cu that has nanosized structures. Figure 1e simulates the distribution of the electromagnetic field (EM) hotspot with the Cu islands larger than the 60 nm - Au particles. What is the EM field distribution for the Au on smaller (<60 nm diameter) Cu Island? How about the enhancement factor compared to Au on >60 nm Cu island as depicted in Figure 1e?
2. The roughened Cu surface results in inhomogeneous charge distribution at the Cu surface, which should lead to inhomogeneous SEI growth. In other words, SEI chemistry should be different on the EM hotspot than on the plain surface - authors should clarify this point
3. Figure S3 shows that the surface is rough and there are several places showing SHIN Au particles

stack together to form multiple layer particles. Figure 1e and Figures S3 b and c only describes the EM distribution of those single-layer SHIN Au on Cu- how to confirm that the Raman laser (assuming 1 μm spot size) is on single-layer SHIN Au particles on Cu, but not on the spots where those SHIN Au stacked together during the in situ Raman experiment?

B. The authors should clarify it: Evidence in Fig s20a shows the existence of the Cu_2O but in all finite element field simulations there is no Cu_2O layer on Cu surface.

C. The authors should comment on how the lithiation of Cu and Cu_2O affects the electromagnetic field distribution and enhancement factor.

D. The authors should explain if the SiO_2 shell in SHIN Au particles goes through the lithiation process and if so, whether the SEI composition is the same as on the Li surface.

E. The authors should explain how they prevent the reaction of Teflon with Li (used as casing for the working electrode) with Li during the Li plating stage.

F. Only simulated enhancement factors (EF) are shown. What is the EF value of Cu nanoparticles and that of the SHIN particles? Are the EF values the same for these nanostructured surrounded by electrolytes compared to when they are surrounded by SEI? Does the EF change when SEI compositions change?

Response to Reviewers

We would like to express our appreciation for the reviewers' positive response and valuable comments on our manuscript. And we are grateful to the opportunity given to clarify the concerns raised in the reviewer's report. We hope that our additional experiments and careful replies and revisions adequately address the reviewers' comments and strengthen our revised manuscript. Our responses to the comments are listed point-by-point as follows, and the corresponding changes made in the revised manuscript are highlighted in yellow.

Reviewer #1

The work presented is a systematic study of the use of SHINERS and SERS to follow SEI formation on Cu substrates. A key outcome is the demonstration of discriminating depth and region of the SEI - by design and combination of the substrates.

Insights from the spectroscopy led to authors demonstrating direct improvement to the electrochemical properties of lithium plating and stripping on Cu – by identifying a less stable SEI formed on the copper and providing a method to address this.

Overall this is a well carried out study with important insights on the nature of the SEI formed upon Cu/Li.

Response: We are grateful for the reviewer's general overview, and we also greatly appreciate the reviewer's positive comments on the quality and importance of our work.

The authors give a good level of citation of the past literature. They may wish to consider adding citations to earlier pioneering Raman studies on lithium metal - particularly work from Rey et al *Electrochimica Acta* 43, 1998, 1539-1544 and Naudin et al *Journal of Power Sources* 124, 2003 518-525 (paper ends with a pertinent remark - "However, the sensitivity of normal Raman is rather limited and SERS effects would be very helpful to characterize the SEI") 20 years later this is beginning to be realised, and the SHINERS study on lithium metal from Galloway et al. *Faraday Discuss.*, 2017, 205, 469-490.

On page 7 the authors state that "such a sequential formation and reconstruction of SEIs are overlooked in past research". A recent study from Unwin and co-workers (*Angewandte Chemie International Edition* 61, e202207184 2022) shows that monitoring SEI formation on Si via SHINERS shows the dynamic nature of SEI formation as well. Agree overlooked, but studies

are appearing that begin to identify the changing nature of SEIs which this work adds to.

Response: We appreciate the reviewer's valuable comments on references. We also apologize for missing some of the pioneering works, and thank the reviewer for bringing them to our attention. We have carefully studied these papers and totally agree with the reviewer that these studies are of far-reaching significance in identifying and understanding the dynamic nature of SEIs. We have now included these references as Ref. 42, Ref. 43, Ref. 52, and Ref. 53 in the main text of the revised manuscript.

Reviewer #2

The manuscript titled “Resolving the nanostructure and chemistry of solid-electrolyte interphase in lithium metal batteries by depth-sensitive plasmon-enhanced Raman spectroscopy” proposed a new method of probing the solid/electrolyte interphase on Li anode by using shell-isolated Au nanoparticles SERS media. The major finding is that the SEI evolution on the Cu and on plated Li can be sequentially explored. Insights learned from this method benefit optimization of the electrochemical cycling of the Li metal battery to generate a more stable SEI to promote cell performance. Overall, it is nice work and should attract the broad interest of the researchers working in the Li-ion battery field. Before it can be published in Nature Communications, the following comments should be properly addressed:

Response: We deeply appreciate the reviewer for the praise related to the quality of our work and its broad interest to the research community working in the Li-ion battery field. We also appreciate the constructive comments which we respond to below.

A. Regarding the principle of DS-PERS described in Figure 1:

1. Early formation of SEI is probed by the roughened surface of Cu that has nanosized structures. Figure 1e simulates the distribution of the electromagnetic field (EM) hotspot with the Cu islands larger than the 60 nm - Au particles. What is the EM field distribution for the Au on smaller (<60 nm diameter) Cu Island? How about the enhancement factor compared to Au on >60 nm Cu island as depicted in Figure 1e?

Response: We thank the reviewer for this comment. The structure selected for modeling the localized surface plasmon (LSP) enhancement of integrated Cu-SHINs substrate in this work was based on the experimental result that the average size of Cu islands is larger than ~60 nm

SHINs (as shown in Supplementary Fig. 1), along with the operability of simulation. While we agree with the reviewer that the nanostructured Cu surface obtained by electrochemical oxidation-reduction cycle method potentially has a relatively broad size distribution including those smaller than 60 nm. To address the reviewer's question, we have modelled the EM field distribution for the integrated Cu-SHINs substrate with smaller Cu island (57 nm in diameter) using the finite element method, as shown in Fig. R1a. In this configuration, three types of hotspots are still formed in the junctions of Cu-to-Cu (I), Cu-to-SHIN (II), and SHIN-to-SHIN (III) under 785 nm excitation, the same as the Cu-SHINs substrate with larger Cu island (Fig. R1b). And the average plasmon enhancement factor (EF) of Cu-SHINs substrate with smaller Cu island is two orders of magnitude lower than that of Cu-SHINs substrate with larger Cu island, but still reaches up to 8 orders of magnitude. Despite the two orders of magnitude decrease in EF it is still sufficiently strong for the detection of SEI components, as it falls into the standard thresholds of EF ($>10^7$) for ultrasensitive LSP-active substrate for detecting interfacial species (*Mater. Today* 2012, 15, 16–25; *Nat. Rev. Mater.* 2016, 1, 16021). Therefore, regardless of the size of Cu island, the integrated Cu-SHINs substrate can guarantee a tremendous sensitivity to the monitoring of SEI formation and evolution.

We have included the related simulation result in Supplementary Fig. 3 and incorporated the corresponding discussions to Note S3 in the revised Supplementary Information to better reflect the enhancement ability of integrated Cu-SHINs substrate.

Fig. R1 | **a,b**, Numerical simulation of the Raman enhancement distribution of the integrated Cu-SHINs substrate with smaller (a) and larger (b) Cu islands, respectively.

2. The roughened Cu surface results in inhomogeneous charge distribution at the Cu surface, which should lead to inhomogeneous SEI growth. In other words, SEI chemistry should be different on the EM hotspot than on the plain surface – authors should clarify this point.

Response: We thank the reviewer for pointing out an important issue. We agree with the reviewer that the SEI chemistry on roughened Cu surface could change somehow. The reasons include inhomogeneous charge distribution as well as higher surface area of the nanostructured morphology, the latter causing a significantly more consumption of electrolyte and consequently reformation of SEI.

Regarding the surface area effect, it mostly occurs when the surface area of the nanostructured Cu surface is tens to hundreds of times of a planar Cu surface (*Nat. Commun.* 2015, 6, 8058; *Adv. Energy Mater.* 2018, 8, 1800266; *J. Mater. Chem. A*, 2021, 9, 24963–24970; *Chem. Eng. J.* 2022, 450, 138384). In our work, the real surface area of the roughened Cu was examined by using Pb underpotential deposition (UPD) as shown in Fig. R2, and the calibrated surface area is only nearly 2 times of the geometric surface area of planar Cu (1 mm in diameter). In order to ascertain whether such a roughened Cu surface in our system affects the electrolyte reduction kinetics (i.e., SEI formation), we first performed CV measurement that are highly sensitive to electrolyte decomposition pathways. As presented in Fig. R3, it is clearly seen that the electrochemical behavior of electrolyte on roughened Cu is identical to that on the planar Cu, which indicates that the SEI formation process is not significantly impacted by the roughened Cu surface with slightly increased surface area, at least on a macroscopic scale.

Fig. R2 | CV measurement of Pb UPD on the roughened Cu in the electrolyte of 1 mM $\text{Pb}(\text{ClO}_4)_2$ + 10 mM HClO_4 + 1 mM HCl at scan rate of 10 mV s^{-1} . The real surface area of roughened Cu is calibrated according to charge density under the corresponding Pb UPD peaks.

Fig. R3 | CV curves of the planar Cu (upper) and roughened Cu (lower) electrodes in the electrolyte of 1 M LiTFSI/DME-DOL at scan rate of 50 mV s^{-1} .

Regarding the hotspot effect (i.e., charge distribution), LSPs excited by plasmonic nanostructures are indeed possible to mediate chemical reactions, which is also one of the research interests of our group. However, thus far, LSPs have demonstrated the capability to mediate only limited solar energy-related reactions, such as water splitting and CO_2 reduction (*Matter* 2020, 3, 42–56). Meanwhile, the benefits of the LSPs to these reactions is limited primarily to enhancing conversion under the specific conditions, and the efficiencies of many such reactions remain quite low (*Nat. Rev. Chem.* 2018, 2, 216–230). In order to inspect whether the LSPs excited in our roughened Cu-SHINs substrate affect the SEI formation, we performed some further experiments as follows. On the one hand, the composition of SEIs formed in different regions of roughened Cu-SHINs substrate with and without continuous laser irradiation during SEI formation process were compared. As shown in Fig. R4, the two regions exhibit similar spectroscopic characters after SEI formation, indicating the SEIs formed in both regions have the similar composition and structure, and the excited LSPs do not alter the electrolyte's decomposition pathway. On the other hand, we introduced LSP-active Au@SiO_2 SHINs onto the planar Cu surface to enhance the Raman signals of SEI grown on it and compared the results with those obtained on the roughened Cu-SHINs substrate. Fig. R5 displays the Raman spectra obtained in both cases. It can be seen that the reduction products from electrolyte are not detected on planar Cu-SHINs substrate until the potential reached 1.2 V. The intensities of these Raman spectra are weaker than those from roughened Cu-SHINs substrate, and this can be attributed to the relatively lower detection sensitivity of planar Cu-SHINs substrate to the initial stage of SEI formation. However, the planar Cu-SHINs presents the similar spectroscopic characters upon further negative potential excursion, which suggests that the composition and structure SEI formed on planar Cu is same as

that formed on the roughened one. These results reveal that the hotspot effect on SEI chemistry is also negligible in our system.

Fig. R4 | Comparison of Raman spectra of different regions on Cu-SHINs substrate at 1.0 V (vs. Li/Li^+) in DOL-based electrolyte. Point A: SEI formation under continuous laser irradiation; Point B: SEI formation without continuous laser irradiation.

Fig. R5 | In-situ Raman spectra showing the SEIs formation on roughened Cu-SHINs (a) and planar-SHINs (b) substrates, respectively, in DOL-based electrolyte.

The above-present results clearly demonstrate that the roughened Cu surface in our system does not seem to significantly affect the SEI chemistry. Nevertheless, we totally agree with the reviewer that the use of more uniform SERS-active substrate with appropriate surface area and homogeneous charge distribution would avoid excessive interferences and thus enable more accurate characterization of SEI formation and evolution. Further work employing a uniform

substrate with specific nanostructures in close proximity to each other, such as Cu nanoparticle array, is underway in our group.

We have incorporated the corresponding discussions to Note S2 in the revised Supplementary Information.

3. Figure S3 shows that the surface is rough and there are several places showing SHIN Au particles stack together to form multiple layer particles. Figure 1e and Figures S3 b and c only describes the EM distribution of those single-layer SHIN Au on Cu- how to confirm that the Raman laser (assuming 1 μm spot size) is on single-layer SHIN Au particles on Cu, but not on the spots where those SHIN Au stacked together during the in situ Raman experiment?

Response: We thank the reviewer for this comment. Indeed, we cannot completely preclude SHINs from stacking together to form multiple layers somewhere on the substrate, although most of SHINs are obviously spread as a monolayer over the substrate as shown in Supplementary Fig. 3. It is very hard to identify the proportion of multiple-layer SHINs included in one laser spot (ca. 2 μm in diameter). In order to clarify the influence of multi-layered SHINs on the LSP enhancement, we carried out a theoretical simulation by using finite element method. Fig. R6 illustrates the calculated EM distribution on Cu-SHINs substrate with multi-layered SHINs, taking a bilayer SHINs as an example. It clearly shows that the introduction of extra SHINs in the upper layer barely interferes with the original strength and distribution of hotspots on the bottom layer and thus the detection of SEIs. However, it should be noted that the thickness of SEIs formed in the electrolyte systems presented in this work is typically below 50 nm. Therefore, a monolayer SHINs on nanostructured Cu would not be buried during SEI formation and can effectively serve as the Raman signal amplifier. Moving forward, we envision that for those systems with thicker SEIs (>50 nm), deliberate assembly of multi-layered SHINs on nanostructured Cu and/or reasonable optimization of the size of plasmonic metal core is essential to meet the on-demand tracking of SEIs. We are thankful that this comment leads us to come to a much more comprehensive understanding of working principle of DS-PERS.

Fig. R6 | Numerical simulation of the EM distribution of the integrated Cu-SHINs substrate with bilayer SHINs.

B. The authors should clarify it: Evidence in Fig s20a shows the existence of the Cu₂O but in all finite element field simulations there is no Cu₂O layer on Cu surface.

Response: We thank the reviewer for this comment. The formation of minor Cu oxides on Cu surface is practically unavoidable during the preparation of nanostructured Cu electrode, despite meticulous operation have been applied, because of the common knowledge that Cu is easily oxidized even in the presence of a small amount of oxygen. However, these oxides are completely reduced to metallic Cu upon negative potential excursion prior to electrolyte reduction (i.e., SEI formation), as evidenced by the absent of doublet intensity (Supplementary Fig. 20a in the original Supplementary Information), which is also confirmed in previous literature (*Electrochim. Acta* 1985, 30, 1687–1692; *J. Phys. Chem. C* 2021, 125, 16719–16732). Therefore, we simulated the enhancement capability of nanostructured Cu without considering the effect of Cu oxides, and such processing is also consistent with those reported in other studies that use the similar type of Cu substrates (*J. Am. Chem. Soc.* 2019, 141, 12192–12196; *Energy Environ. Sci.*, 2022, 15, 3968–3977).

C. The authors should comment on how the lithiation of Cu and Cu₂O affects the electromagnetic field distribution and enhancement factor.

Response: We thank the reviewer for this comment. As discussed in the response to comment B, the minor Cu₂O species have already been completely reduced to metallic Cu with negative shifting of potential and thus has little effect on the EF of the nanostructured Cu substrate. Regarding the lithiation of Cu, it is generally believed that Cu does not undergo lithiation processes but only Li deposition on its surface (*Nat. Energy* 2016, 1, 16010; *Prog. Mater. Sci.* 2022, 130, 100996). In the original manuscript, the influence of Li deposition on the electromagnetic field distribution in the integrated Cu-SHINs substrate has already been sated

(see Page 5, Line 8 in the original main text and Supplementary Note 5). The key findings are that after Li deposition, the metallic Li replaces Cu to serve as new Raman signal amplifier, and Au@SiO₂ SHINs floating on Li deposits can generate effective hotspots located at the junctions of Li-to-Li, Li-to-SHIN and SHIN-to-SHIN and thus still maintain the high enhancement factor of up to 10¹⁰ (Fig. 1e and Supplementary Fig. 7c,d in the original Supplementary Information), which enables ultrasensitive detection of Raman signals of SEI components.

D. The authors should explain if the SiO₂ shell in SHIN Au particles goes through the lithiation process and if so, whether the SEI composition is the same as on the Li surface.

Response: We thank the reviewer for highlighting this point. Indeed, there is a school of thought considering SiO_x (0 < x ≤ 2) as a promising family of anode materials for lithium-ion batteries. However, regarding SiO₂ that was once thought to be electrochemically inactive toward Li due to its intrinsic poor ionic conductivity and sluggish Li⁺ transport properties, the controversies still exist about whether there is a reversible lithiation/de-lithiation processes of SiO₂ and what the reaction mechanism between Li and SiO₂ is (*J. Power Sources*, 1999, 81–82, 362–367; *Adv. Funct. Mater.*, 2007, 17, 1765–1774; *Chem. Soc. Rev.*, 2019, 48, 285–309; *Adv. Energy Mater.* 2020, 10, 2001826; *Int. J. Miner. Metall. Mater.*, 2022, 876–895). As literature studies have suggested, the lithiation behavior of SiO₂ highly depends on its size, crystallinity, morphology, and oxygen content. (*Adv. Mater.* 2021, 33, 2004577; *Adv. Energy Mater.* 2022, 12, 2202342). To inspect whether the SiO₂ shell in our system goes through the lithiation process, we have first performed galvanostatic discharge measurements on Cu and Cu-SHIN substrates in DOL-based electrolyte, respectively. Obviously, the potential-time profiles of the two substrates in Fig. R7 show similar characteristics, in which the region above 0 V corresponds to the electrochemical decomposition of the electrolyte to form the SEI. No discharge plateau related to the lithiation reaction of SiO₂ can be observed on Cu-SHIN substrate. Then, the discharged Cu-SHIN substrate was washed with DME to remove residual electrolyte and put into DME solution for 1 h of ultrasonic cleanout, and finally the fallen Au@SiO₂ SHINs were collected for TEM imaging. As shown in Fig. R8, the thickness of SiO₂ shell are virtually unchanged after discharge, compared to the initial SiO₂ shell with thickness of ~2 nm, and no volumetric strain induced by lithiation can be detected. The above-presented experiments suggest that the SiO₂ shell in our system does not undergo the lithiation process (or if it takes place, it is too inconspicuous to interfere with the experiments). However, we agree with reviewer that SiO₂ shell may have a potential risk of lithiation, and more inert shell should be developed to avoid

the potential reaction with Li. Actually, the related work, such as the preparation of Au/Ag nanoparticles with a more inert shell (such as ZrO_2 and HfO_2), is underway in our group.

We have added the related discussions into Supplementary Fig. 2 of the revised Supplementary Information.

Fig. R7 | Potential profiles of galvanostatic Li deposition on bare Cu and Cu-SHINs substrates, respectively, at -0.1 mA cm^{-2} .

Fig. R8 | TEM images Au@SiO₂ nanoparticles before and after discharge. Scale bar, 50 nm.

E. The authors should explain how they prevent the reaction of Teflon with Li (used as casing for the working electrode) with Li during the Li plating stage.

Response: We thank the reviewer for this comment. Actually, we have tried to encapsulate the Cu electrode into various casings composed of different types of materials, including Teflon, PEEK and epoxy. And we found that the Teflon casing is more stable in our system than the other two type of casings.

Fig. R9 | CV curves of the Cu encapsulated in PEEK (a) and Tefflon (b), and assembled in coin cell (c) in the DOL-based ether electrolyte.

Specifically, for the epoxy casing, it undergoes either a physical or chemical change when in contact with electrolytes, especially ether-based one. As to the PEEK casing, there are no macroscopic physical changes when exposed to electrolytes. Nevertheless, a series of unclear cathodic/anodic current responses emerge during CV experiment with particularly ether-based electrolyte when the scanning potential goes more negative than the potential of Li UPD (Fig. R9a). These responses are not observed when using the routine coin cell configuration (Fig. R9c) and may indicate an (electro-)chemical reaction between PEEK and electrolyte with involvement of metallic Li. While for the Tefflon casings, a normal current response is obtained during CV experiment in ether-based electrolyte over a potential range that included the emergence of Li OPD (Fig. R9b), the same as that obtained in routine coin cell configuration. This suggests that the reaction of Tefflon with metallic Li and electrolyte may be too weak to interfere with the experiments. Moreover, when we perform Raman experiments, the $\sim 2 \mu\text{m}$ laser spot is generally focused to the central region of the Cu electrode (1 mm in diameter), and the spot will not irradiate the edge of the electrode, i.e., the joint between the Cu electrode and the Tefflon casing (as shown in Supplementary Fig. 8 in the original Supplementary Information).

Therefore, even if the Teflon reacts with deposited Li during the Li plating stage, it does not affect the Raman experiment significantly. Actually, there are many literature using Teflon to encapsulate the metal electrodes and/or manufacture electrolytic cells, and some even press Li chips directly onto Teflon products (*Nat. Mater.*, 2014, 13, 961–969; *ACS Energy Lett.*, 2018, 1, 14–19; *J. Am. Chem. Soc.* 2019, 141, 18612–18623). Yet it seems none of these literature studies have reported noticeable side reactions of Teflon and Li. However, we agree with the reviewer that further efforts are needed to prevent the potential reaction of Teflon with Li in the future, such as replacing the Cu electrode with a Cu foil without casing for Raman experiments, which will also be more convenient to perform operando characterization under conditions close to the practical batteries.

F. Only simulated enhancement factors (EF) are shown. What is the EF value of Cu nanoparticles and that of the SHIN particles? Are the EF values the same for these nanostructured surrounded by electrolytes compared to when they are surrounded by SEI? Does the EF change when SEI compositions change?

Response: We thank the reviewer for this comment. The EF values of three types of hotspots located on the Cu-SHINs integrated substrate in electrolyte are given below in Fig. R10.

Fig. R10 | EF values of three types of hotspots located on the Cu-SHINs integrated substrate before and after SEI formation.

These values were calculated to be $10^9 \sim 10^{10}$ magnitude, demonstrating a strong capability of such configurations for ultrasensitive detection of surface species. After SEI formation, the calculated EF value of each hotspot remains to be $10^9 \sim 10^{10}$ magnitude (Fig. R10), which is similar to that of initial substrate before SEI formation. As already discussed in the original manuscript (Page 5, Line 1) and Supplementary Information (Supplementary Note 4), the SEIs are composed primarily of various inorganic species, such as LiF, Li₂O, LiOH, etc., as well as organic species like lithium alkyl carbonates, and these components absorb light weakly and

have very similar refractive indexes and/or dielectric constants, which are close to that of electrolyte (*Handbook of Inorganic Chemicals, McGraw-Hill, 2003; J. Phys. Chem. 1996, 100, 3089–3101*). Therefore, the formation of SEI and/or the change in SEI components are not expected to cause significant changes in the EF values of these hotspots, which is an underlying principle that enables the DS-PERS method to effectively monitor the formation and evolution of SEIs.

We have added these results as Supplementary Table 1 in the revised Supplementary Information.

REVIEWERS' COMMENTS

Reviewer #2 (Remarks to the Author):

The authors have done an exhaustive job in addressing all comments, and they have performed excellent work in revising the manuscript